# Understanding deep neural networks through the lens of their non-linearity

## Abstract

The remarkable success of deep neural networks (DNN) is often attributed to their high expressive power and their ability to approximate functions of arbitrary complexity. Indeed, DNNs are highly non-linear models, and activation functions introduced into them are largely responsible for this. While many works studied the expressive power of DNNs through the lens of their approximation capabilities, quantifying the non-linearity of DNNs or of individual activation functions remains an open problem. In this paper, we propose the first theoretically sound solution to track non-linearity propagation in deep neural networks with a specific focus on computer vision applications. Our proposed affinity score allows us to gain insights into the inner workings of a wide range of different architectures and learning paradigms. We provide extensive experimental results that highlight the practical utility of the proposed affinity score and its potential for long-reaching applications.

## 1 Introduction

What makes deep neural networks so powerful? This question has been studied extensively since the very inception of the field and along several different paths. First contributions in this direction aimed to show that neural networks are universal approximators (Barron, 1994; Kurt & Hornik, 1991; Cybenko, 1989) and can fit any function to the desirable accuracy. Such results first required considered NNs to be of infinite width or depth: both constraints were finally relaxed to show that such property also holds for NNs in a finite regime (Hanin, 2017; Lu et al., 2017). Another line of research aimed at quantifying the expressive power of DNNs to understand their recent successes. First such contributions considered specifically piece-wise linear activation functions such as the famous ReLU or less frequently used hard tanh. The idea behind these works was to see DNNs as piece-wise linear functions (Montúfar et al., 2014) or as linear functions for a fixed set of activation patterns (Raghu et al., 2017b). In this case, the expressiveness measure is equal either to the number of the linear regions or the number of activation patterns respectively. The former idea has also been extended to the case of convolutional neural networks(Xiong et al., 2020).

**Limitations** Undeniably, activation functions are at the core of understanding the expressive power of DNNs as they are largely responsible for making DNNs such powerful function approximators. Yet, previous works overlook this fundamental property of activation functions while concentrating on new proposals making the optimization of DNNs more efficient. The ReLUs (Glorot et al., 2011) were proposed as an alternative to saturating activation functions such as sigmoid, leaky ReLUs (Maas et al., 2013) tackled the dying ReLU problem, and smoother activation functions, such as GELU (Hendrycks & Gimpel, 2016) and SiLU (Elfwing et al., 2018), were proposed to have a better gating in DNNs using thresholding based on the value of input rather than its sign. But beyond this, do we know how much non-linearity such functions introduce depending on the layer and the architecture in which they are used? Is there a way to compare different activation functions between themselves? These questions are of prime importance as the choice of activation function has a strong impact on the dynamics of the learning process in practice. Our contributions toward answering these questions are as follows:

1. We propose a theoretically sound tool, called the affinity score, that measures the non-linearity of a given transformation using optimal transport (OT) theory.

2. We apply the proposed affinity score to a wide range of popular DNNs used in computer vision: from Alexnet to vision transformers (ViT). We show how the different developments in deep learning can be understood through the prism of non-linearity propagation.

3. We perform extensive tests applying the affinity score to compare different computational approaches both in terms of scale and in terms of the architectural choices they implement. This gives us a unique viewpoint for better understanding DNNs in practice.

The rest of the paper is organized as follows. We start by presenting the relevant background knowledge on OT in Section 2. Then, we introduce the affinity score together with its different theoretical properties in Section 3. Section 4 presents experimental evaluations on a wide range of popular convolutional neural networks. Finally, we conclude in Section 5.

## 2 BACKGROUND

**Optimal Transport** Let $(X, d)$ be a metric space equipped with a lower semi-continuous *cost function* $c : X \times X \to \mathbb{R}_{\geq 0}$, e.g the Euclidean distance $c(x, y) = \|x - y\|$. Then, the Kantorovich formulation of the OT problem between two probability measures $\mu, \nu \in \mathcal{P}(X)$ is given by

$$\text{OT}_c(\mu, \nu) = \min_{\gamma \in \text{ADM}(\mu, \nu)} \mathbb{E}_\gamma[c], \tag{1}$$

where $\text{ADM}(\mu, \nu)$ is the set of joint probabilities with marginals $\mu$ and $\nu$, and $\mathbb{E}_\nu[f]$ denotes the expected value of $f$ under $\nu$. The optimal $\gamma$ minimizing equation 1 is called the *OT plan*. Denote by $\mathcal{L}(X)$ the law of a random variable $X$. Then, the OT problem extends to random variables $X, Y$ and we write $\text{OT}_c(X, Y)$ meaning $\text{OT}_c(\mathcal{L}(X), \mathcal{L}(Y))$.

Assuming that either of the considered measures is *absolutely continuous*, then the Kantorovich problem is equivalent to the *Monge problem*

$$\text{OT}_c(\mu, \nu) = \min_{T : T_{\#}\mu = \nu} \mathbb{E}_{X \sim \mu}[c(X, T(X))], \tag{2}$$

where the unique minimizing $T$ is called the *OT map*, and $T_{\#}\mu$ denotes the *push-forward measure*, which is equivalent to the *law* of $T(X)$, where $X \sim \mu$.

**Wasserstein distance** Let $X$ be a random variable over $\mathbb{R}^d$ satisfying $\mathbb{E}[\|X - x_0\|^2] < \infty$ for some $x_0 \in \mathbb{R}^d$, and thus for any $x \in \mathbb{R}^d$. We denote this class of random variables by $\mathcal{P}_2(\mathbb{R}^d)$. Then, the 2-Wasserstein distance $W_2$ between $X, Y \in \mathcal{P}_2(\mathbb{R}^d)$ is defined as

$$W_2(X, Y) = \text{OT}_{\|x-y\|^2}(X, Y)^{\frac{1}{2}}, \tag{3}$$

## 3 AFFINITY SCORE

### 3.1 DEFINITION

Defining the non-linearity of a transformation from observed samples is difficult as the latter doesn't admit any commonly accepted quantitative equivalent. One tangible solution to this problem is to measure non-linearity as *lack of linearity*. For this idea to prove fruitful, it requires a notion of a *perfect* lack of linearity acting as an upper bound. Moreover, we would like to expect such a measure to be applicable without any prior assumptions regarding the statistical nature of the observed quantities. Below, we present a quantitative measure of linearity with the desired properties.

**Identifiability** Our first step towards building the affinity score is to ensure that we are capable of identifying the situation when two random variables are linked through an affine transformation. To show that such an identifiability condition can be satisfied with OT, we first recall the following classic result from the literature characterizing the OT maps.

**Theorem 3.1** (Smith & Knott (1987)). *Let $X \in \mathcal{P}_2(\mathbb{R}^d)$, $T(x) = \nabla\phi(x)$ for a convex function $\phi$ with $T(X) \in \mathcal{P}_2(\mathbb{R}^d)$. Then, $T$ is the unique optimal OT map between $\mu$ and $T_{\#}\mu$.*

Using this theorem about the uniqueness of OT maps expressed as gradients of convex functions, we can immediately prove the following result (all proofs can be found in the Appendix A):

**Corollary 3.2.** *Without loss of generality, let $X, Y \in \mathcal{P}_2(\mathbb{R}^d)$ be centered, and such that $Y = TX$, where $T$ is a positive semi-definite linear transformation. Then, $T$ is the OT map from $X$ to $Y$.*

Whenever two random variables are linked through an affine transformation, the solution to the OT problem between them is exactly the affine transformation we are looking for.

**Characterization** We now know that solving the OT problem between two random variables linked through an affine transformation allows us to identify the latter. Now, we seek to understand whether this solution can be characterized more explicitly. For this, we prove the following theorem stating that when $X$ and $Y$ differ by an affine transformation, $W_2(X, Y)$ can be computed in closed-form using their normal approximations *no matter how complicated $X$ and $Y$ are as distributions.*

**Theorem 3.3.** *Let $X, Y \in \mathcal{P}_2(\mathbb{R}^d)$ be centered and $Y = TX$ for a positive definite matrix $T$. Let $N_X \sim \mathcal{N}(\mu(X), \Sigma(X))$ and $N_Y \sim \mathcal{N}(\mu(Y), \Sigma(Y))$ be their normal approximations where $\mu$ and $\Sigma$ denote mean and covariance, respectively. Then, $W_2(N_X, N_Y) = W_2(X, Y)$ and $T = T_{\mathrm{aff}}$, where $T_{\mathrm{aff}}$ is the OT map between $N_X$ and $N_Y$ and can be calculated in closed-form*

$$T_{\mathrm{aff}}(x) = Ax + b, \quad A = \Sigma(Y)^{\frac{1}{2}} \left( \Sigma(Y)^{\frac{1}{2}} \Sigma(X) \Sigma(Y)^{\frac{1}{2}} \right)^{-\frac{1}{2}} \Sigma(Y)^{\frac{1}{2}},$$
$$b = \mu(Y) - A\mu(X). \tag{4}$$

**Upper bound** If $X$ and $Y$ differ by a more complicated transformation than an affine one, how can we characterize this lack of linearity? One important step in this direction is given by the famous Gelbrich bound, formalized by the means of the following theorem:

**Theorem 3.4** (Gelbrich bound (Gelbrich, 1990)). *Let $X, Y \in \mathcal{P}_2(\mathbb{R}^d)$ and let $N_X, N_Y$ be their normal approximations. Then, $W_2(N_X, N_Y) \leq W_2(X, Y)$.*

This upper bound provides a first intuition of why OT can be a great tool for measuring the non-linearity: the cost of the affine map solving the OT problem on the left-hand side increases when the map becomes non-linear. We now upper bound the difference between $W_2(N_X, N_Y)$ and $W_2(X, Y)$, two quantities that coincide *only* for random variables linked through an affine transformation.

**Proposition 3.5.** *Let $X, Y \in \mathcal{P}_2(\mathbb{R}^d)$ and $N_X, N_Y$ be their normal approximations. Then,*

*1.* $|W_2(N_X, N_Y) - W_2(X, Y)| \leq \frac{2 \operatorname{Tr}\left[ (\Sigma(X)\Sigma(Y))^{\frac{1}{2}} \right]}{\sqrt{\operatorname{Tr}[\Sigma(X)] + \operatorname{Tr}[\Sigma(Y)]}}.$

*2. For $T_{\mathrm{aff}}$ as in (4), $W_2(T_{\mathrm{aff}}X, Y) \leq \sqrt{2} \operatorname{Tr}[\Sigma(Y)]^{\frac{1}{2}}$.*

The bound given in Proposition 3.5 lets us define the following *affinity score*

$$\rho_{\mathrm{aff}}(X, Y) = 1 - \frac{W_2(T_{\mathrm{aff}}X, Y)}{\sqrt{2} \operatorname{Tr}[\Sigma(Y)]^{\frac{1}{2}}}, \tag{5}$$

describing how much $Y$ differs from being a positive-definite affine transformation of $X$. In particular, $0 \leq \rho_{\mathrm{aff}}(X, Y) \leq 1$, and $\rho_{\mathrm{aff}}(X, Y) = 1$ if and only if $X$ and $Y$ differ by an affine transformation. Similarly, $\rho_{\mathrm{aff}}(X, Y) = 0$ when the OT plan between $T_{\mathrm{aff}}X$ and $Y$ is the independent distribution $\mathcal{L}(X) \otimes \mathcal{L}(Y)$.

**Preliminary results** Before exploring the application of our proposal to multi-layered DNNs, we first illustrate it on a single pass of the data through the following activation functions: Sigmoid, ReLU (Glorot et al., 2011), and GELU (Hendrycks & Gimpel, 2016) and cover other popular choices in Appendix B. As the non-linearity of activation functions depends on the domain of their input, we fix 20 points in their domain equally spread in $[-20, 20]$ interval. We use these points as means $\{m_i\}_{i=1}^{20}$ of Gaussian distributions from which we sample 1000 points in $\mathbb{R}^{300}$ with standard deviation (std) $\sigma$ taking values in $[2, 1, 0.5, 0.25, 0.1, 0.01]$. Each sample denoted by $X_{m_i}^{\sigma_j}$ is then passed through the activation function act $\in \{\text{sigmoid}, \text{ReLU}, \text{GELU}\}$ to obtain $\rho_{\mathrm{aff}}^{m_i, \sigma_j} := \rho_{\mathrm{aff}}(X_{m_i}^{\sigma_j}, \text{act}(X_{m_i}^{\sigma_j}))$. Larger std values make it more likely to draw samples that are closer to the region where the studied activation functions become non-linear. We present the obtained results in Figure 1 where each of 20 boxplots showcases median($\rho_{\mathrm{aff}}^{m_i, \sigma_\cdot}$) values with 50% confidence intervals and whiskers covering the whole range of obtained values across all $\sigma_j$.

Figure 1: Median affinity scores of Sigmoid, ReLU and GELU (**black line**) obtained across random draws from Gaussian distribution with a sliding mean and varying stds used as their input. Whiskers of boxplots show the whole range of values obtained for each mean across all stds. The baseline value is the affinity score obtained for a sample covering the whole interval. *The ranges and extreme values of each activation function over its subdomain are indicative of its non-linearity limits.*

This plot allows us to derive several important conclusions. First, we note that the domain of the activation function has a drastic effect on its non-linearity! In this sense, we observe that each activation function can be characterized by 1) the lowest values of its non-linearity obtained for some subdomain of the considered interval and 2) the width of the interval in which it maintains its non-linearity. We note that in terms of 1) both GELU and ReLU may attain affinity scores that are close to 0, which is not the case for Sigmoid. For 2), we observe that the non-linearity of Sigmoid and GELU is maintained in a wide range, while for ReLU it is rather narrow.

## 3.2 IMPLEMENTATION

We now turn our attention to the application of the affinity score to the analysis of deep neural networks. In this work, we concentrate mainly on convolutional neural networks (convnets) for which we possess a large repository of pre-trained models made available as part of *torchvision* (maintainers & contributors, 2016). In addition to convnets, we also consider transformer architectures with a specific focus on the non-linearity present in their MLP blocks.

**Non-linearity signature** We define a convnet N as a composition of layers $F_i$ where each layer $F_i$ is a function taking as input a tensor $X_i \in \mathbb{R}^{h_i \times w_i \times c_i}$ (for instance, an image of size $224 \times 224 \times 3$ for $i = 1$) and outputting a tensor $Y_i \in \mathbb{R}^{h_{i+1} \times w_{i+1} \times c_{i+1}}$ used as an input of the following layer $F_{i+1}$. This defines $N = F_L \odot ... \odot F_i ... \odot F_1(X_1) = \bigodot_{k=1,...,L} F_k(X_1)$ where $\odot$ stands for a composition. For each $F_i$, which is also a composition of operators often including a convolution, a pooling, and an activation function, we particularly concentrate on the latter.

We now present the definition of a non-linearity signature of a network N. Below, we abuse the compositional structure of $F_i$ and see it as an ordered sequence of functions.

**Definition 3.1.** *Let* $N = \bigodot_{k=1,...,L} F_k(X_1)$ *be a neural network. Define by ACT a finite set of common activation functions such that* $ACT := \{\sigma | \sigma : \mathbb{R}^{h \times w \times c} \to \mathbb{R}^{h \times w \times c}\}$. *Let* $r$ *be a function such that* $r : \mathbb{R}^{h \times w \times c} \to \mathbb{R}^c$. *Then, the non-linearity signature of N is defined as follows:*

$$\rho_{\text{aff}}(N) = (\rho_{\text{aff}}(act(r(\cdot)), r(\cdot))), \forall act \in F_i \cap ACT, \quad i = 1, \ldots, L.$$

Non-linearity signature associates to each network N a vector of affinity scores calculated over the inputs and outputs of all activation functions encountered across its layers. We now discuss the role of the function $r$ used in its definition.

**Dimensionality reduction** Manipulating 4-order tensors is computationally prohibitive and thus we need to find an appropriate lossless function $r$ to facilitate this task. One possible choice for $r$ may be a vectorization operator that flattens each tensor into a vector. In practice, however, such flattening still leads to very high-dimensional data representations. In our work, we propose to use averaging over the spatial dimensions to get a suitable representation of the manipulated tensors. In Figure 2 (left), we show that the affinity score is robust with respect to such an averaging scheme and maintains the same values as its flattened counterpart.

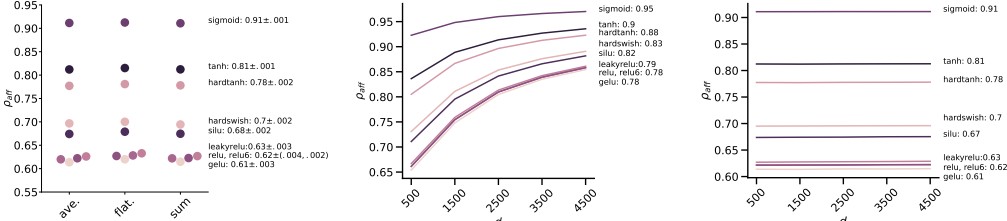

Figure 2: **(left)** Affinity score is robust to the dimensionality reduction both when using averaging and summation over the spatial dimensions; **(middle)** When $d > n$, sample covariance matrix estimation leads to a lack of robustness in the estimation of the affinity score; **(right)** Shrinkage of the covariance matrix leads to constant values of the affinity scores with increasing $d$. In the Appendix C, we show the robustness of the affinity score to batch size and different seeds.

**Computational considerations**   The non-linearity signature requires calculating the affinity score over "wide" matrices. Indeed, after the reduction step is applied to a batch of $n$ tensors of size $h \times w \times c$, we end up with matrices of size $n \times c$ where $n$ may be much smaller than $c$. This is also the case when input tensors are 2D when the batch size is smaller than the dimensionality of the embedding space. To obtain a well-defined estimate of the covariance matrix in this case, we use a known tool from the statistics literature called Ledoit-Wolfe shrinkage (Ledoit & Wolf, 2004). In Figure 2 (right), we show that shrinkage allows us to obtain a stable estimate of the affinity scores that remain constant in all regimes.

### 3.3   RELATED WORK

**Layer-wise similarity analysis of DNNs**   A line of work that can be distantly related to our main proposal is that of quantifying the similarity of the hidden layers of the DNNs as proposed Raghu et al. (2017a) and Kornblith et al. (2019) (see (Davari et al., 2023) for a complete survey of the subsequent works). Raghu et al. (2017a) extracts activation patterns of the hidden layers in the DNNs and use CCA on the singular vectors extracted from them to measure how similar the two layers are. Their analysis brings many interesting insights regarding the learning dynamics of the different convnets, although they do not discuss the non-linearity propagation in the convnets, nor do they propose a way to measure it. Kornblith et al. (2019) proposed to use a normalized Frobenius inner product between kernel matrices calculated on the extracted activations of the hidden layers and argued that such a similarity measure is more meaningful than that proposed by Raghu et al. (2017a).

**Impact of activation functions**   Dubey et al. (2022) provides the most comprehensive survey on the activation functions used in DNNs. Their work briefly discusses the non-linearity of the different activation functions suggesting that piecewise linear activation functions with more linear components are more non-linear (e.g., ReLU vs. ReLU6). Our work provides a more in-depth understanding of different activation functions when analyzed both in isolation and as part of a DNN. In particular, we show that the non-linearity of an activation function is ill-defined without specifying its domain and the architecture in which they are used. Hayou et al. (2019) show theoretically that smooth versions of ReLU allow for more efficient information propagation in DNNs with a positive impact on their performance. We show that such functions are more non-linear and exhibit wider regions of high non-linearity.

**Non-linearity measure**   The only work similar to ours in spirit is the paper by Philipp (2021) proposing the non-linearity coefficient in order to predict the train and test error of DNNs. Their coefficient is defined as a square root of the Jacobian of the neural network calculated with respect to its input, multiplied by the covariance matrix of the Jacobian, and normalized by the covariance matrix of the input. The presence of the Jacobian in it calls for the differentiability assumption making its application to most of the neural networks with ReLU non-linearity impossible as is. The authors didn't provide any implementation of their coefficient and we were not able to find any other study reporting the reproduced results from this work.

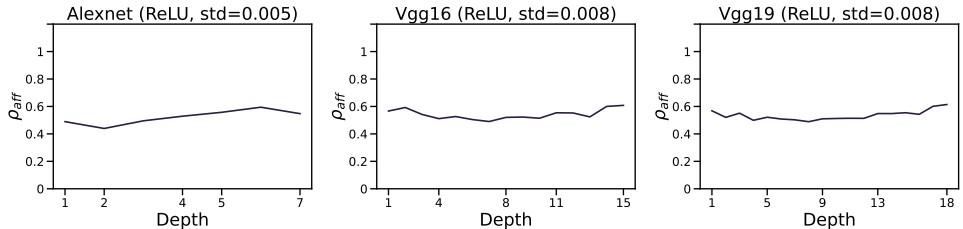

Figure 3: Early convnets: Alexnet and its successors VGG16 and VGG19.

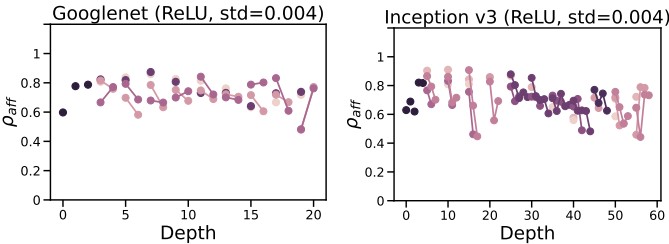

Figure 4: Going deeper with Inception module: Googlenet and Inception v3 architectures. Overlayed points are parallel branches in the inception modules.

## 4 EXPERIMENTAL EVALUATIONS

We now present our main experimental results using the DNNs pre-trained on Imagenet-1K dataset that are made publicly available in torchvision repository. The non-linearity signature of convnets is calculated by passing batches of size 512 through the pre-trained models for the entirety of the Imagenet-1K validation set with a total of 50000 images (see Appendix E for results on CIFAR 10/100 datasets, and randomly generated data). In all plots portraying the non-linearity signature, different color codes stand for *distinct* activation functions appearing *repeatedly* in the considered architecture (for instance, every first ReLU in a residual block of a Resnet). We also indicate the mean standard deviation of the affinity scores over batches in the title to indicate just how stable it is.

### 4.1 HISTORY OF CONVNETS: FROM ALEXNET TO TRANSFORMERS

**Alexnet** In the last decade we have witnessed remarkable progress in DNNs with far reaching applications. But if one were to single out one paper that can be seen as a major catalyst for the scientific interest behind this progress, then it would undoubtedly be the Alexnet paper of Krizhevsky et al. (2012). Beyond winning the Imagenet 2012 Challenge by a large margin, Alexnet marked a new frontier in visual feature learning starting the era of dominance of convnets that lasts until today.

We thus start our presentation with this seminal architecture in Figure 3 and use it as a baseline to introduce the evolution of the field seen since then. Our first observation in this regard is rather striking: **the non-linearities across ReLU activations in all of Alexnet's 8 layers remain stable!** Does it mean that the success of Alexnet is explained by its ability to push ReLU non-linearity to its limits?! Our further analysis will show a surprising finding suggesting that higher non-linearity – lower affinity score – doesn't necessarily correlate with better performance.

**VGG** First illustration of this is the VGG network (Simonyan & Zisserman, 2015): an improvement over Alexnet that introduced a first, by the standards of that time, a very deep convolutional neural network with roughly twice the number of layers of its predecessor. **The non-linearity signature of VGG16 reveals tiny, yet observable, variations in the non-linearity propagation** with increasing depth and, slightly lower overall non-linearity values. We attribute this to the decreased size of the convolutional filters (3x3 vs. 7x7) rather than depth as the latter behavior is observed across VGG11, VGG13, VGG16, and VGG19 architectures (see Appendix D).

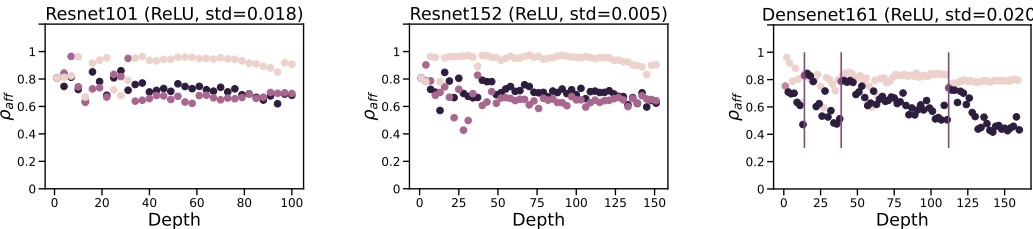

Figure 5: First convnets with skip connections: Resnet101, Resnet152 and Densenet161. Vertical lines of Densenet stand for the transition layers between its blocks.

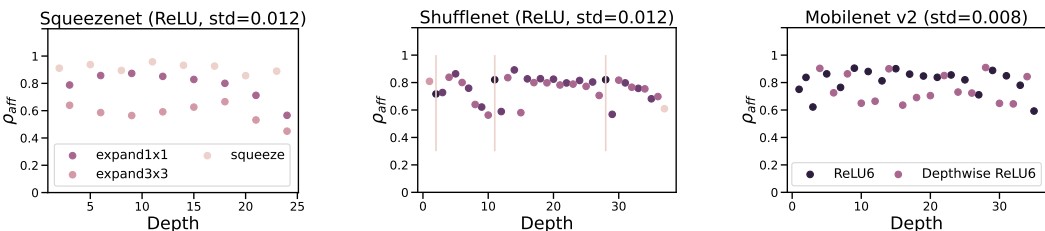

Figure 6: Striving for efficiency: Squeezenet (2016), Shufflenet (2017), Mobilenet (2018) networks.

**Googlenet and Inception** Our next stop in this journey through time is the Googlenet architecture (Szegedy et al., 2014): a first model to consider learning features at different scales in parallel within the so-called inception modules. Figure 4 shows **how much variability the inception modules introduce: affinity scores vary between 0.6 and 0.9**. Despite being almost 20 times smaller than VGG16, the accuracy of Googlenet on Imagenet remains comparable, suggesting that increasing and varying the linearity is a way to have high accuracy with a limited computational complexity compared to predecessors. The inception module used in Googlenet has found its further application in another state-of-the-art model, a winner of the Imagenet competition in 2016, termed Inception v3 (Szegedy et al., 2016). We present the non-linearity signature of this model in Figure 4. We observe what now becomes a clear trend: **the non-linearity signature of Inception v3 is shaken up to an even higher degree** pushing it toward being more linear in some hidden layers. When comparing this behavior with Alexnet, we note just how far we are from it.

**Resnet and Densenet** We now enter a different realm: that of learning very deep models with over 100 layers. This breakthrough was made possible by Resnet (He et al., 2016) thanks to the use of skip connections. Figure 5 presents the non-linearity signature of Resnet152 (Resnet18, Resnet34, Resnet50 and Resnet101 are in Appendix D). We can see that **Resnets achieve the same effect of "shaking up" the non-linearity but in a different, and arguably, simpler way**. Indeed, the activation after the skip connection exhibits affinity scores close to 1, while the activations in the hidden layers remain much lower. We further compare the non-linearity signature of the same network with and without residual connections in the Appendix F. Further down this road, Densenet (Huang et al., 2017) sought to train deeper architectures using dense connections that connect each layer to all previous layers and not just to the one that precedes it. Figure 5 shows the comparison between the two. We can see that **Densenet161 is slightly more non-linear than Resnet152**, although the two bear a striking similarity: they both have an activation function that maintains the non-linearity low with increasing depth. Additionally, **transition layers in Densenet act as linearizers** and allow it to reset the non-linearity propagation in the network by reducing the feature map size.

**Towards model efficiency** ML field has witnessed a significant effort in optimizing DNNs efficiency. This effort was dictated by its high practical importance: large models couldn't run on mobile devices limiting their further potential impact. To this end, several models suitable for on-device use were proposed as early as 2015. In Figure 6 we present three famous models that were part of this quest for efficiency: Squeezenet (Iandola et al., 2016), Shufflenet (Zhang et al., 2017) and Mobilenet (Howard et al., 2017). Squeezenet has a peculiar non-linearity signature showing that squeeze modules act as transition modules in Densenet and reduce the non-linearity. Apart from that, all three

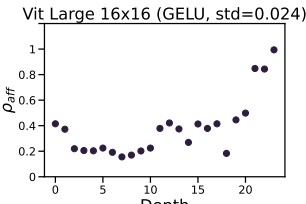 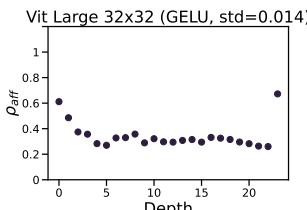 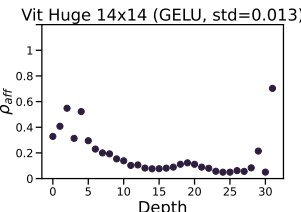

Figure 7: ViTs: Large ViT with 16x16 and 32x32 patch sizes and Huge ViT.

models show a clear trend of non-linearity "shaking", achieved either through squeeze-and-excite modules, shuffling of channels, or inverted residual bottlenecks.

**Vision transformers**    We now come to the last chapter of our journey, the modern times dominated by vision transformers (ViT) Dosovitskiy et al. (2021). In Figure 7, we show that the original **ViTs** (Large with 16x16 and 32x32 patch sizes, and Huge with 14x14 patches) **are highly non-linear** to the degree yet unseen. Interestingly, the patch size affect the non-linearity propagation in a non-trivial way: for 16x16 size a model is more non-linear in the early layers, while gradually becoming more and more linear later, while 32x32 patch size leads to a plateau in the hidden layers of MLP blocks, with a steep change toward linearity only in the final layer. These results remind us of Figure 1 where we highlighted the importance of the domain of the activation function on its non-linearity. We hypothesize that attention modules in ViT act as a focusing lens and output the embeddings in the domain where the activation function is the most non-linear.

### 4.2    Affinity score does not correlate strongly with other metrics

We now aim to understand whether other intuitive ways to measure the non-linearity correlate with the affinity score. We consider linear CKA (Kornblith et al., 2019) presented in Section 3.4 [1], the average change in SPARSITY and ENTROPY before and after the application of the activation function as well as the Frobenius NORM between the input and output of the activation functions. Finally, we consider a measure of linearity given by the $R^2$ score between the linear model fitted on the input and the output of the activation function. From Figure 8 (upper left), we see that **no other criterion consistently correlates with the affinity score** across 33 architectures used in our test. We note that CKA points in the right direction for some architectures (Densenets are a notable example), although for others it correlates negatively. We believe that this experiment provides evidence of the affinity score being rather unique in its ability to capture the non-linearity propagation in DNNs.

### 4.3    Affinity score defines a meaningful notion of similarity between convnets

In this experiment, we show that different convnets can be identified, on a group level, based on their non-linearity signature. To this end, we calculate a matrix of pairwise distances for non-linearity signatures of 33 models considered above using Dynamic Time Warping (DTW) (Sakoe & Chiba, 1978) distance. DTW allows us to handle the non-linearity signatures of different sizes. In Figure 8 (lower left-right) we present the result of the hierarchical clustering performed on this matrix. We can see that **all semantically close architectures are grouped together without any apparent outliers**. We also note just how different are the two biggest ViTs from the rest of the models present in this study. We provide a similar comparison for self-supervised models in Appendix G.

### 4.4    Affinity score is predictive of the Imagenet performance

We now show how the performance on Imagenet dataset correlates with the non-linearity signature of different models. As different architectures behave very differently, we split them in 3 semantically meaningful groups: 1) Traditional models (Alexnet, VGGs) and residual models (Resnets and Densenets); 2) size- and design-efficient networks such as Squeezenet, Shufflenet, and ConvNext; and

---

[1]Linear CKA performs on par with its kernelized counterpart (Kornblith et al., 2019) and almost all subsequent papers used the linear version Davari et al. (2023), which explains our choice as well.

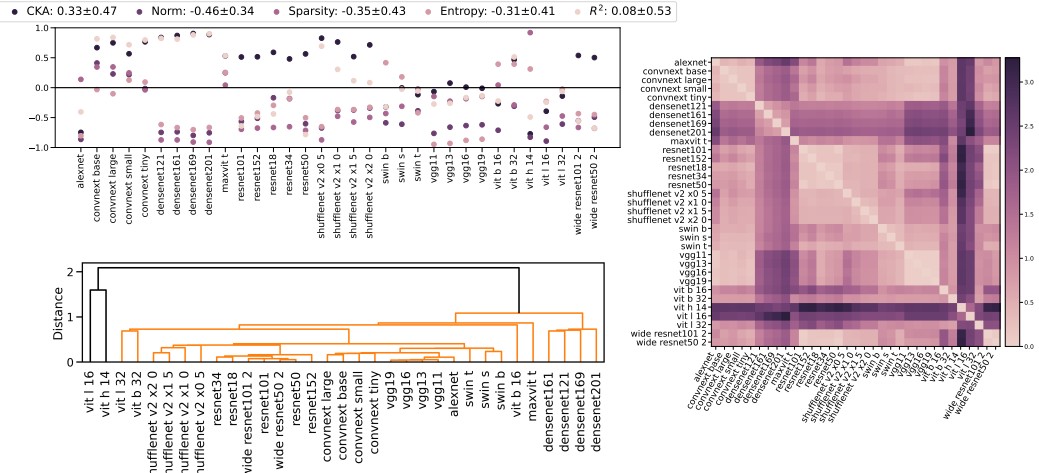

Figure 8: **Affinity score is a unique non-linearity measure**. **(upper-left)** Correlations between the affinity score and other metrics; **(lower left)** Clustering of the architectures using the pairwise DTW **(right)** distances between their non-linearity signatures.

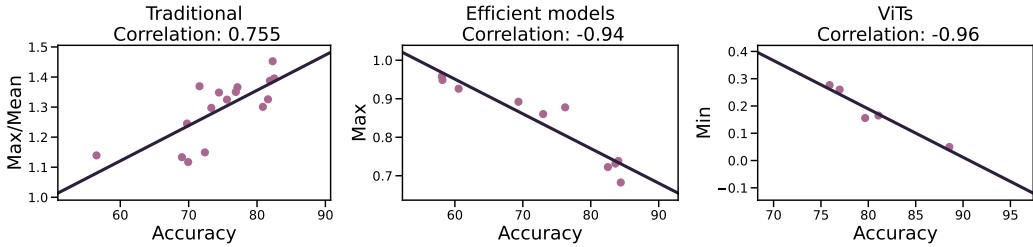

Figure 9: Information extracted from the non-linearity signature correlates strongly with Imagenet Acc@1 performance.

3) ViTs. As the non-linearity signature is a vector, we seek to understand what 1D statistic extracted from it correlates most – either positively or negatively – with their Imagenet performance. Without going too deep into feature engineering, we choose as candidates the average, standard deviation, minimum, maximum, and the ratio of the maximum to the mean. The latter choice is dictated by our observation regarding a clear positive impact of having varying degrees of non-linearity in the network. In Figure 9, we present the results of this comparison highlighting the chosen statistic achieving maximum absolute correlation with Acc@1 accuracy of the model on Imagenet.

We note several surprising findings. First, we see that the non-linearity signature is not only predictive of the Imagenet performance of a model but also tells us which particular aspect of it is optimized by each family of DNNs to improve the performance (see Appendix I for comparison with other common complexity metrics). Second, size-efficient architectures seem to be penalized for having highly linear activation functions as the maximum affinity score across their layers correlates negatively with the final accuracy. Finally, in the case of transformers, we observe a clear trend toward the results improving when the model becomes more and more non-linear.

## 5   CONCLUSION

In this paper, we presented a metric called *affinity score* that quantifies the non-linearity of a given transformation. We further proposed a way to apply it to the analysis of the deep convolutional neural networks by defining their non-linearity signature as a vector of affinity scores associated with the activation functions used in them. We showed that non-linearity scores bring insights into the inner workings of many different architectures starting from Alexnet and ending with ViTs.

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
