## A    PROOFS OF MAIN THEORETICAL RESULTS

In this section, we provide proofs of the main theoretical results from the paper.

**Corollary 3.2.** Without loss of generality, let $X, Y \in \mathcal{P}_2(\mathbb{R}^d)$ be centered, and such that $Y = TX$, where $T$ is a positive semi-definite linear transformation. Then, $T$ is the OT map from $X$ to $Y$.

*Proof.* We first proof that we can consider centered distributions without loss of generality. To this end, we note that

$$W_2^2(X, Y) = W_2^2(X - \mathbb{E}[X], Y - \mathbb{E}[Y]) + \|\mathbb{E}[X] - \mathbb{E}[Y]\|^2, \tag{6}$$

implying that splitting the 2-Wasserstein distance into two independent terms concerning the $L^2$ distance between the means and the 2-Wasserstein distance between the centered measures.

Furthermore, if we have an OT map $T'$ between $X - \mathbb{E}[X]$ and $Y - \mathbb{E}[Y]$, then

$$T(x) = T'(x - \mathbb{E}[X]) + \mathbb{E}[Y], \tag{7}$$

is the OT map between $X$ and $Y$.

To prove the statement of the Corollary, we now need to apply Theorem 3.1 to the convex $\phi(x) = x^T T x$, where $T$ is positive semi-definite. $\qquad \square$

**Theorem 3.3.** Let $X, Y \in \mathcal{P}_2(\mathbb{R}^d)$ be centered and $Y = TX$ for a positive definite matrix $T$. Let $N_X \sim \mathcal{N}(\mu(X), \Sigma(X))$ and $N_Y \sim \mathcal{N}(\mu(Y), \Sigma(Y))$ be their normal approximations where $\mu$ and $\Sigma$ denote mean and covariance, respectively. Then, $W_2(N_X, N_Y) = W_2(X, Y)$ and $T = T_{\text{aff}}$, where $T_{\text{aff}}$ is the OT map between $N_X$ and $N_Y$ and can be calculated in closed-form

$$T_{\text{aff}}(x) = Ax + b, \quad A = \Sigma(Y)^{\frac{1}{2}} \left( \Sigma(Y)^{\frac{1}{2}} \Sigma(X) \Sigma(Y)^{\frac{1}{2}} \right)^{-\frac{1}{2}} \Sigma(Y)^{\frac{1}{2}},$$
$$b = \mu(Y) - A\mu(X). \tag{8}$$

*Proof.* Corollary 3.2 states that $T$ is an OT map, and

$$\Sigma(TN_X) = T\Sigma(X)T = \Sigma(Y).$$

Therefore, $TN_X = N_Y$, and by Theorem 3.1, $T$ is the OT map between $N_X$ and $N_Y$. Finally, we compute

$$\begin{aligned}
W_2^2(N_X, N_Y) &= \text{Tr}[\Sigma(X)] + \text{Tr}[T\Sigma(X)T] - 2\,\text{Tr}[T^{\frac{1}{2}}\Sigma(X)T^{\frac{1}{2}}] \\
&= \underset{T:T(X)=Y}{\arg\min} \; \mathbb{E}_X[\|X - T(X)\|^2] \\
&= W_2^2(X, Y).
\end{aligned}$$

$\qquad \square$

**Proposition 3.5.** Let $X, Y \in \mathcal{P}_2(\mathbb{R}^d)$ and $N_X, N_Y$ be their normal approximations. Then,

1. $|W_2(N_X, N_Y) - W_2(X, Y)| \leq \dfrac{2\,\text{Tr}\left[(\Sigma(X)\Sigma(Y))^{\frac{1}{2}}\right]}{\sqrt{\text{Tr}[\Sigma(X)] + \text{Tr}[\Sigma(Y)]}}$.

2. For $T_{\text{aff}}$ as in (4), $W_2(T_{\text{aff}}X, Y) \leq \sqrt{2}\,\text{Tr}\left[\Sigma(Y)\right]^{\frac{1}{2}}$.

*Proof.* By Theorem 3.4, we have $W_2(N_X, N_Y) \leq W_2(X, Y)$. On the other hand,

$$\begin{aligned}
W_2^2(X, Y) &= \min_{\gamma \in \text{ADM}(X,Y)} \int_{\mathbb{R}^d \times \mathbb{R}^d} \|x - y\|^2 d\gamma(x, y) \\
&\leq \int_{\mathbb{R}^d \times \mathbb{R}^d} \left(\|x\|^2 + \|y\|^2\right) d\gamma(x, y) \\
&= \text{Tr}[\Sigma(X)] + \text{Tr}[\Sigma(Y)].
\end{aligned}$$

Combining the above inequalities, we get

$$|W_2(N_X, N_Y) - W_2(X, Y)| \leq \left| \sqrt{\text{Tr}[\Sigma(X)] + \text{Tr}[\Sigma(Y)]} - W_2(N_X, N_Y) \right|.$$

Let $a = \text{Tr}[\Sigma(X)] + \text{Tr}[\Sigma(Y)]$, and so $W_2^2(N_X, N_Y) = a - b$, where $b = 2\,\text{Tr}\left[ (\Sigma(X)\Sigma(Y))^{\frac{1}{2}} \right]$. Then the RHS of can be written as

$$\left| \sqrt{a} - \sqrt{a - b} \right| = \frac{|a - (a - b)|}{\sqrt{a} + \sqrt{a - b}} \leq \frac{b}{\sqrt{a}},$$

where the inequality follows from positivity of $W_2(N_X, N_Y) = \sqrt{a - b}$. Letting $X = T_{\text{aff}} X$ in the obtained bound gives 2). $\qquad\square$

# B   AFFINITY SCORES OF OTHER POPULAR ACTIVATION FUNCTIONS

In this section, we complete Figure 1 by presenting the same plots for 6 more activation functions including ReLU6 (Howard et al., 2017), LeakyReLU (Maas et al., 2013) with a default value of the slope, Tanh, HardTanh, SiLU (Elfwing et al., 2018), and HardSwish (Howard et al., 2019). We use the exact same protocol as in the figure presented in the main paper.

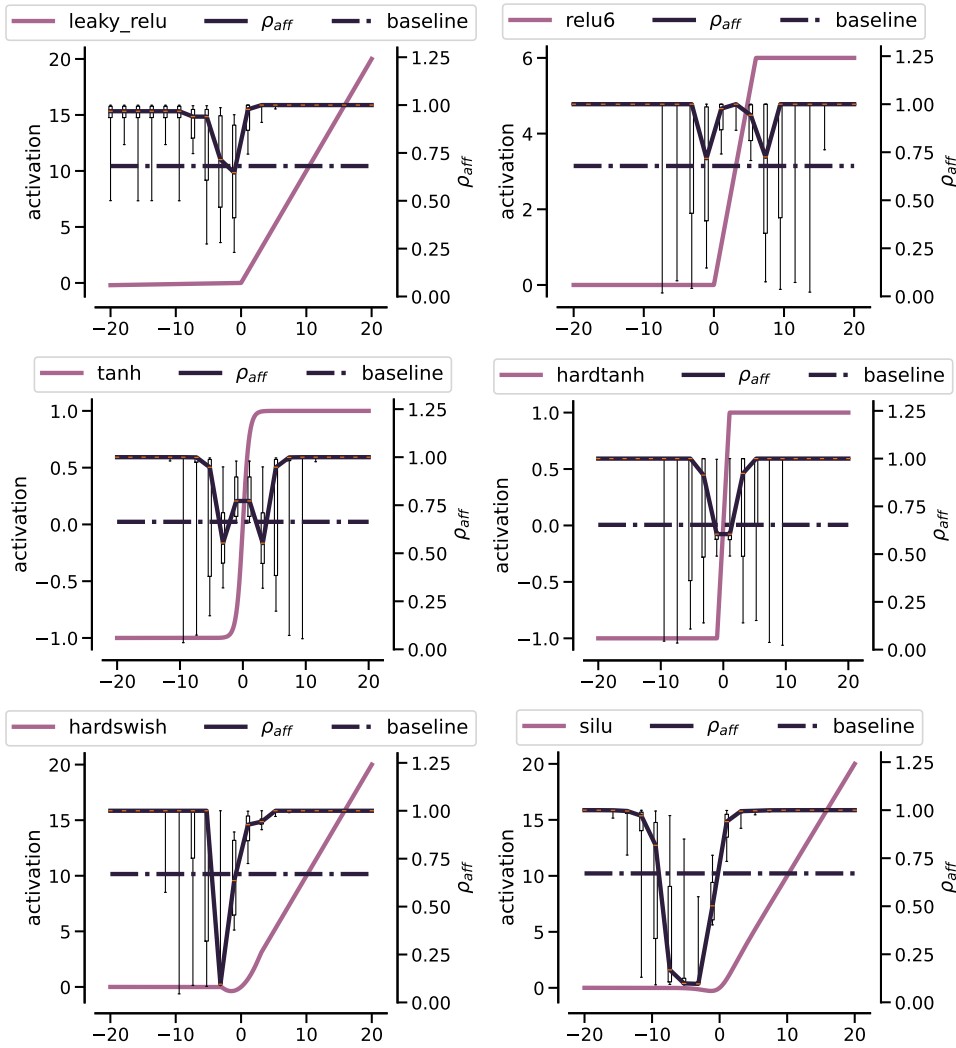

Figure 10: Completing Figure 1 with ReLU6, LeakyReLU with a default value of slope, Tanh, HardTanh, SiLU, and HardSwish.

From Figure 10, we can see a distinct pattern of more modern activation functions, such as SiLU and HardSwish having a stronger non-linearity pattern in large subdomains. We also note that despite having a shape similar to Sigmoid, Tanh may allow for much lower affinity scores. Finally, the variations of ReLU seem to have a very similar shape with LeakyReLU being on average more linear than ReLU and ReLU6.

## C ROBUSTNESS WITH RESPECT TO BATCH SIZE AND RANDOM INITIALIZATION

In this section, we highlight the robustness of the non-linearity signature with respect to the batch size and the random seed used for training. To this end, we concentrate on VGG16 architecture and CIFAR10 dataset to avoid costly Imagenet retraining. In Figure 11, we present the obtained result where the batch size was varied between 128 and 1024 with an increment of 128 (left plot) and when VGG16 model was retrained with seeds varying from 1 to 9 (right plot). The obtained results show that the affinity score is robust to these parameters suggesting that the obtained results are not subject to a strong stochasticity.

Finally, we also show how a non-linearity signature of a VGG16 model looks like at the beginning and in the end of training on Imagenet. We extract its non-linearity signature at initialization when making a feedforward pass over the whole CIFAR10 dataset and compare it to the non-linearity signature obtained in the end. In Figure 12, we can see that at initialization the network's non-linearity signature is increasing reaching almost a perfectly linear pattern in the last layers. Training the network enhances the non-linearity in a non-monotone way.

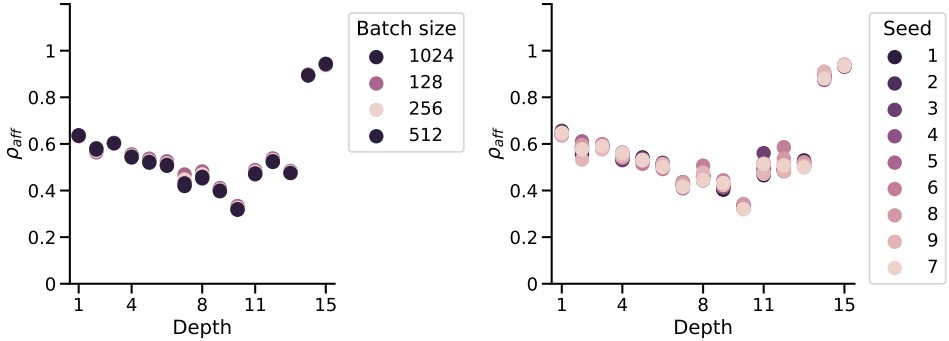

Figure 11: Non-linearity signature of VGG16 on CIFAR10 with a varying batch size (left) and when retrained from 9 different random seeds (right).

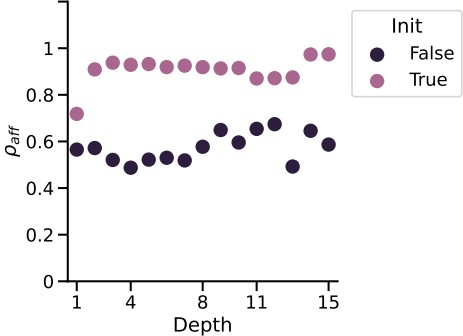

Figure 12: Non-linearity signatures of VGG16 on CIFAR10 in the beginning and end of training on Imagenet.

# D ROLE OF DEPTH FOR VGG AND RESNET

In this section, we explore the role of increasing depth for VGG and Resnet architectures. We consider VGG11, VGG13, VGG16 and VGG19 models in the first case, and Resnet18, Resnet34, Resnet50, Resnet101 and Resnet152. The results are presented in Figure 13 and Figure 14 for VGGs and Resnets, respectively.

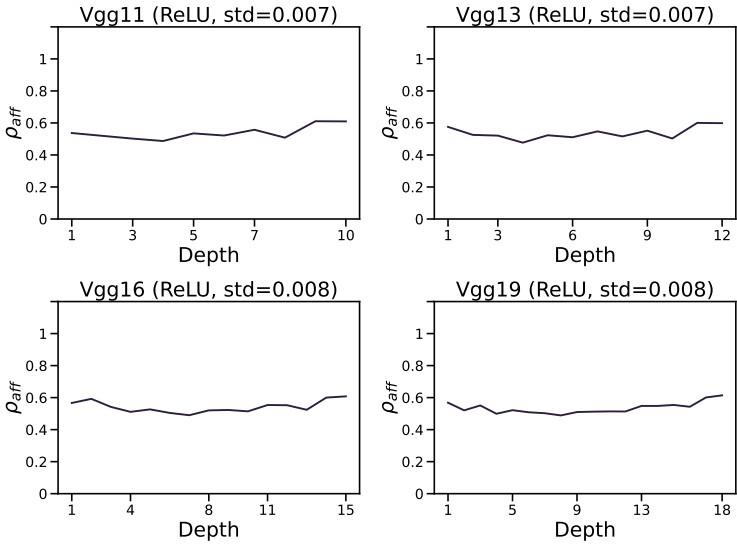

Figure 13: Impact of depth on the non-linearity signature of VGGs.

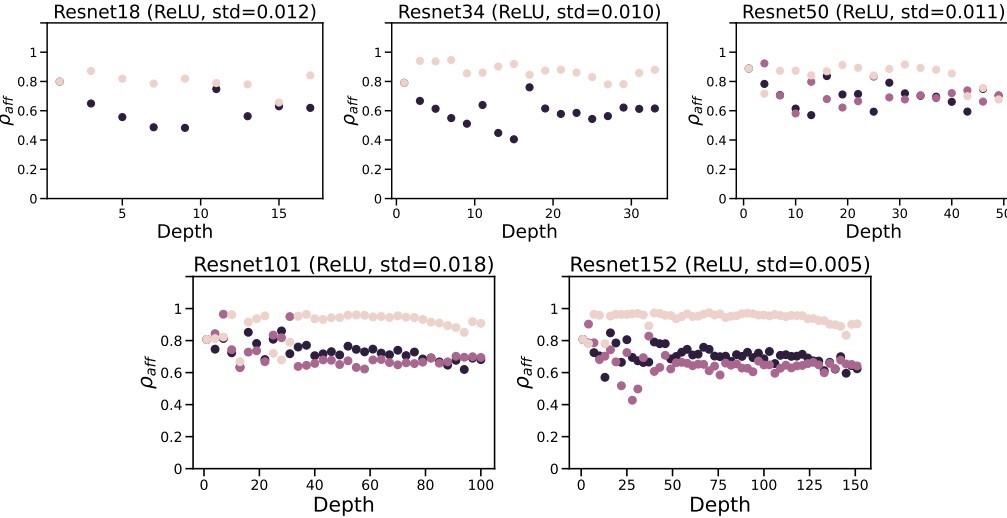

Figure 14: Impact of depth on the non-linearity signature of Resnets.

As mentioned in the main paper, VGGs do not change their non-linearity signature with increasing depth. In the case of Resnets, we can see that the separation between more linear post-residual activations becomes more distinct and approaches 1 for deeper networks.

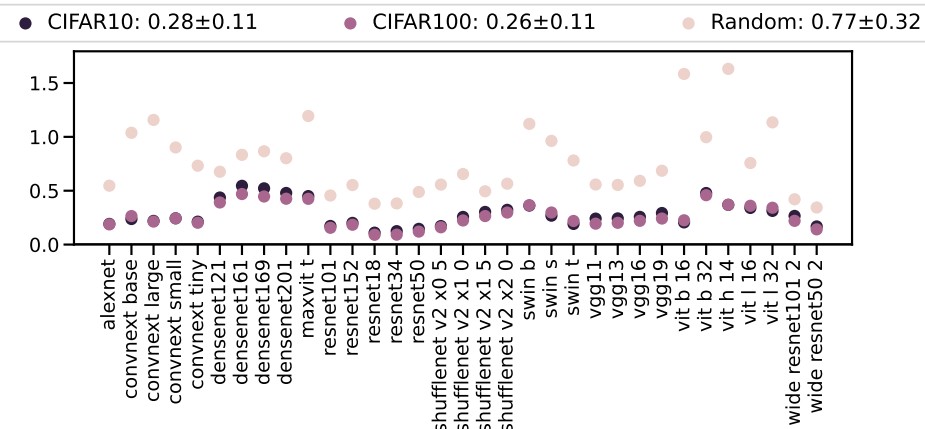

Figure 15: Deviation in terms of the Euclidean distance of the non-linearity signature obtained on CIFAR10, CIFAR100, and Random datasets from the non-linearity signature of the Imagenet dataset.

## E  RESULTS ON MORE DATASETS

Below, we compare the results obtained on CIFAR10, CIFAR100 datasets as well as when the random data tensors are passed through the network. As the number of plots for all chosen 33 models on these datasets will not allow for a meaningful visual analysis, we rather plot the differences – in terms of the DTW distance – between the non-linearity signature of the model on Imagenet dataset with respect to three other datasets. We present the obtained results in Figure 15.

We can see that the overall deviation for CIFAR10 and CIFAR100 remains lower than for Random dataset suggesting that these datasets are semantically closer to Imagenet. This observation, however, is more or less true for different architectures:

# F ROLE OF RESIDUAL CONNECTIONS

We now explore the role of residual connections in Resnet architecture to complete the analysis presented in Figure 5. To this end, we retrain Resnet20 (He et al., 2016) on CIFAR10 with and without skip connections to see how it affects the non-linearity signature of the latter. We present the obtained results in Figure 16. We observe that the affinity scores of a network without skip connections remain roughly in the middle of the non-linearities of the residual network. The average accuracy of the two models over 5 seeds are respectively 90.77±0.24 for Resnet without skip connections and 91.74±0.28 when the latter are used.

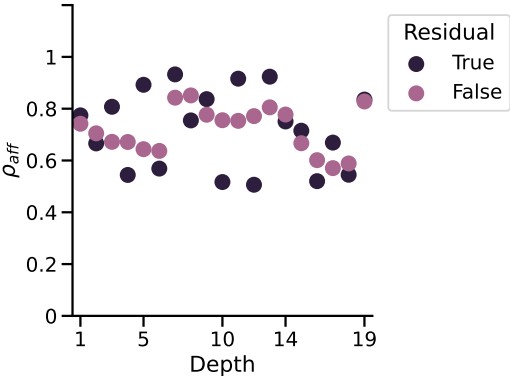

Figure 16: Resnet20 non-linearity signature on CIFAR10 with and without skip connections. We can clearly see that skip connections make the variations of the non-linearity in the network more spread out.

| Criterion | Mean ± std |
|---|---|
| $\rho_{\text{aff}}$ | 0.76±**0.04** |
| Linear CKA | 0.90±0.07 |
| Norm | 448.56±404.61 |
| Sparsity | 0.56±0.16 |
| Entropy | 0.39±0.46 |

Table 1: Robustness of the different criteria when considering the same architectures pre-trained for different tasks. Affinity score achieves the lowest standard deviation suggesting that it is capable of correctly identifying the architecture even when it was trained differently.

## G  RESULTS FOR SELF-SUPERVISED METHODS

In this section, we show that the non-linearity signature of a network remains almost unchanged when considering other pertaining methodologies such as for instance, self-supervised ones. To this end, we use 17 Resnet50 architecture pre-trained on Imagenet within the next 3 families of learning approaches:

1. SwAV (Caron et al., 2020), DINO (Caron et al., 2021), and MoCo (He et al., 2020) that belong to the family of contrastive learning methods with prototypes;

2. Resnet50 (He et al., 2016), Wide Resnet50 (Zagoruyko & Komodakis, 2016), TRex, and TRex* (Sarıyıldız et al., 2023) that are supervised learning approaches;

3. SCE (Denize et al., 2023), Truncated Triplet (Wang et al., 2021), and ReSSL (Zheng et al., 2021) that perform contrastive learning using relational information.

From the dendrogram presented in Figure 17, we can observe that the DTW distances between the non-linearity signatures of all the learning methodologies described above allow us to correctly cluster them into meaningful groups. This is rather striking as the DTW distances between the different instances of the Resnet50 model are rather small in magnitude suggesting that the affinity scores still retain the fact that it is the same model being trained in many different ways.

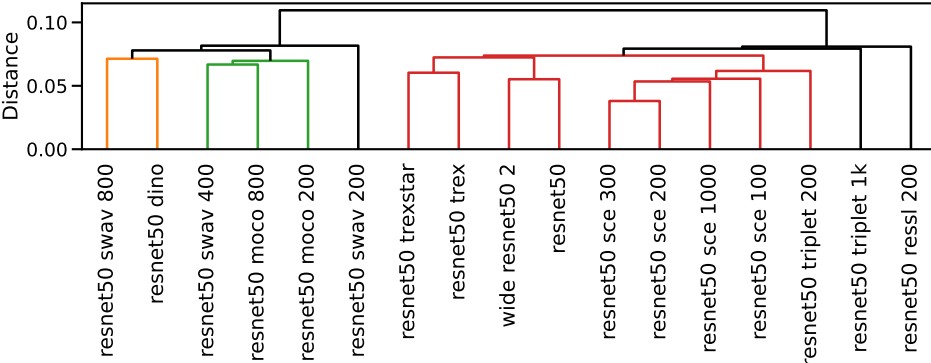

Figure 17: Hierarchical clustering of supervised and self-supervised pre-trained Resnet50 using the DTW distances between their non-linearity signatures.

While providing a fine-grained clustering of different pre-trained models for a given fixed architecture, the average affinity scores over batches remain surprisingly concentrated as shown in Table 1. This hints at the fact that the non-linearity signature is characteristic of architecture but can also be subtly multi-faceted when it comes to its different variations.

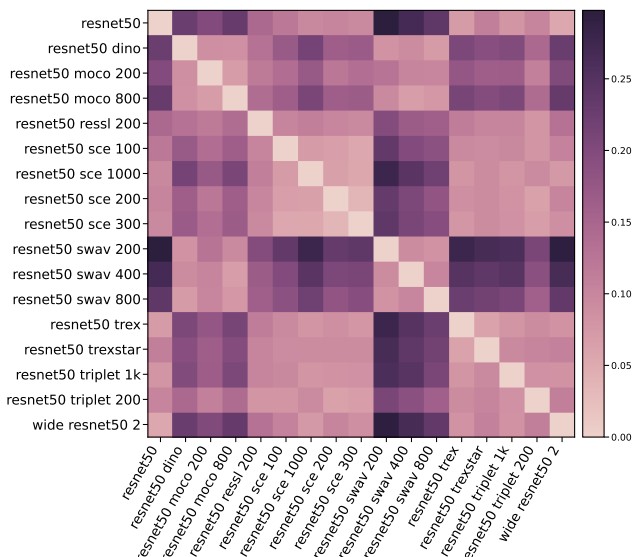

Figure 18: DTW distances associated with the clustering obtained above. We can see distinct clusters as revealed by the dendrogram.

## H    RESULTS ON OTHER OMITTED MODELS

**ViT-like Convnets**    In a recent paper by Liu et al. (Liu et al., 2022), the authors seek to "modernize a standard Convnet (Resnet) towards the design of a hierarchical vision Transformer (Swin) (Cao et al., 2021), without introducing any attention-based modules". In Figure 19, we present the non-linearity signatures of a Resnet50 used as a backbone for Convnext architecture, Convnext resulting from it and the corresponding Swin transformer. Our results highlight the fact that Convnext is indeed a very close imitation of the Swin transformer with both having very close non-linearity signature.

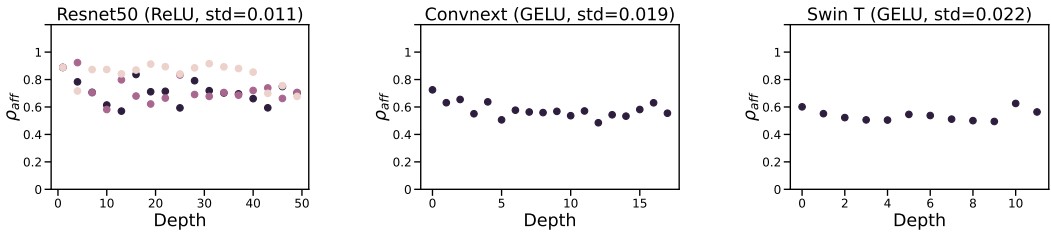

Figure 19: Convnet modernized from Resnet50 backbone looks similar to Swin transformer.

# I COMPARISON WITH OTHER BASELINES IN PREDICTING IMAGENET PERFORMANCE

In this section, we show a drastic difference in the functional relationship that binds the affinity score and other popular metrics related to the DNN complexity to the model's Acc@1 on Imagenet. We consider 3 other metrics retrieved from the models' meta data such as the number of floating point operations (OPS), the DEPTH, and the number of parameters (NB PARAMS). Following the known scaling laws, we use a logarithmic transformation of these variables and look for the best fit characterized by the highest $R^2$ score. We present the obtained results in Figure 20.

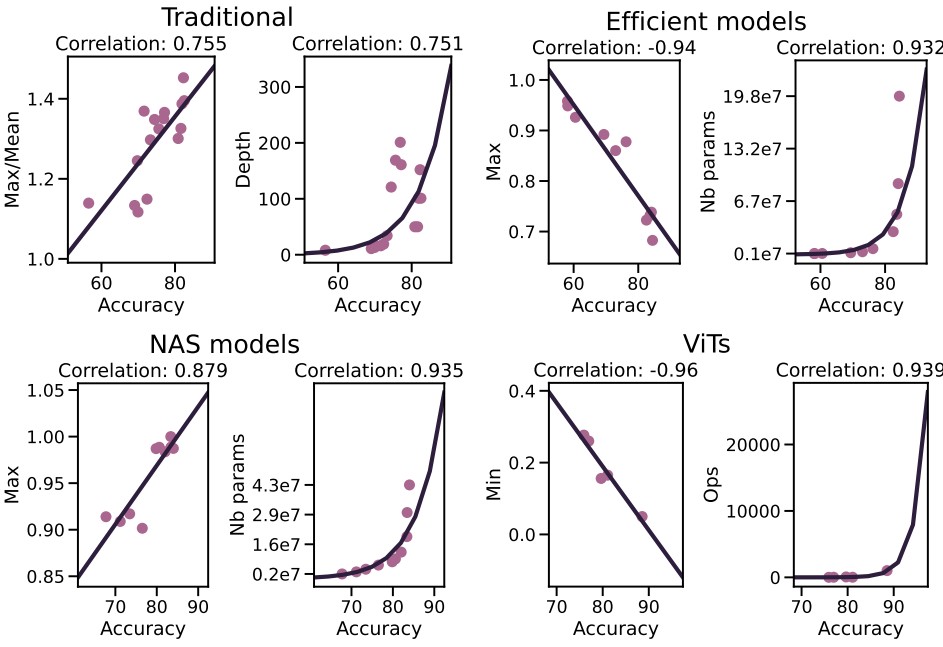

Figure 20: Comparison of the correlation trend between the traditional complexity measures and the non-linearity signature with respect to the Acc@1 on Imagenet.

We note that the logarithmic hypothesis indeed seems to be valid in our case: for every family of models, the usual measures of complexity correlate strongly with the performance following a clear exponential trend. On the contrary, the non-linearity signatures correlated linearly with the Imagenet performance. We also note that for the traditional convnets the best accuracy correlates most with the depth of the model, for NAS and efficient models it is unsurprisingly the number of parameters, while for ViTs it is the number of operations.