# OpenReview forum: "Understanding deep neural networks through the lens of their non-linearity"
_ICLR.cc/2024/Conference — Submitted to ICLR 2024_

### Official Review · Reviewer_Hcf9 · 2023-10-30

**Soundness:** 2 fair
**Presentation:** 3 good
**Contribution:** 2 fair
**Rating:** 5
**Confidence:** 4

**Summary:**

This paper proposes a new metric to measure the non-linearity of activation functions in deep neural networks. Extensive comparative studies have been done for the analysis of different network architectures. The proposed metric is also somehow relevant to the performance of a DNN.

**Strengths:**

+ The topic of analyzing the non-linearity of DNNs is interesting.
+ It seems novel (to me, not an expert) to leverage OT to measure the extent of non-linearity.

**Weaknesses:**

- Def 3.1 only focuses on pointwise non-linear activation functions to measure non-linearity. However, many functions other than the ACT family also introduces non-linearity to the DNN, such as max pooling. How to measure the non-linearity of these functions?
- Def 3.1 considers each activation function independently. How about combining two or more activation functions? Intuitively, by stacking the non-linearities, the overall non-linearity will grow exponentially.
- How to interpret Figure 1? It seems that the non-linearity of sigmoid and gelu are better than relu. Then why does relu become the most widely used activation function?
- About the result in Figure 9. How to interpret the negative correlation between the maximum affinity score and accuracy. It seems contradict with the NAS models results in Figure 20.
- Could the proposed metric bring any new insights (or serve as a regularization term) to help us build better DNNs or design better activation functions?

**Questions:**

see above

---

> ### Author Response · Authors · 2023-11-16
> **Clearing out misunderstandings**
>
> > 1. Def 3.1 only focuses on pointwise non-linear activation functions to measure non-linearity. However, many functions other than the ACT family also introduces non-linearity to the DNN, such as max pooling. How to measure the non-linearity of these functions?
>
> We can measure the non-linearity of any function, point-wise or not, activation or pooling. However, we have chosen the activation functions as they are present in all studied models, often repeatedly. We can include pooling layers as well if we were to extend this work. However, we believe that activation functions, often appearing after conv and pool layers are more interesting in this regard as they effectively capture the non-linearity of what precedes them.
>
> > 2. Def 3.1 considers each activation function independently. How about combining two or more activation functions? Intuitively, by stacking the non-linearities, the overall non-linearity will grow exponentially.
>
> This intuition is actually incorrect as stacking the non-linearity makes them more linear. To see this, we remember that ReLU zeros out all negative elements so that another ReLU on top of its output will become perfectly linear. Nevertheless, we could easily compute affinity scores for combinations of activation functions by considering the combination of functions as a single activation function, but it never happens in practice since there are always other operations between activation functions in neural networks.
>
> > 3. How to interpret Figure 1? It seems that the non-linearity of sigmoid and gelu are better than relu. Then why does relu become the most widely used activation function?
>
> We believe that there is no such a notion as better non-linearity. What it shows is that sigmoid is more linear, and ReLU is highly non-linear only in a very restricted subdomain (interval of the input values). GELU is as non-linear as ReLU but in a larger subdomain making it more stable optimization-wise. We believe that it explains why it is largely used now.
>
> > 4. About the result in Figure 9. How to interpret the negative correlation between the maximum affinity score and accuracy. It seems contradict with the NAS models results in Figure 20.
>
> A negative correlation with the maximum affinity score (maximum over layers) means that a lower maximum affinity score in the efficient model translates to a higher Imagenet top@1 accuracy. NAS models are different from efficient models: it seems that NAS privileges more linear models, while manually designed efficient models are penalized for having too linear activations.
>
> > 5. Could the proposed metric bring any new insights (or serve as a regularization term) to help us build better DNNs or design better activation functions?
>
> We experimenting with this. So far, we managed to successfully generate networks with a predefined non-linearity signature but didn't do any large-scale tests on popular benchmarks.

---

### Official Review · Reviewer_1uvv · 2023-10-31

**Soundness:** 3 good
**Presentation:** 2 fair
**Contribution:** 2 fair
**Rating:** 5
**Confidence:** 3

**Summary:**

This paper proposes a theoretically sound solution to track non-linearity propagation in deep neural networks. Specifically, this method measures the nonlinearity of a given transformation using optimal transport (OT) theory. More critically, the authors investigate the practical utility of the proposed affinity score and apply the proposed affinity score to a wide range of popular DNNs.

**Strengths:**

1. This paper develops a new method to track non-linearity propagation in deep neural networks.
2. This paper proposes the affinity score to evaluate the non-linearity and apply it to diverse architectures.

**Weaknesses:**

1. The authors mention that "consider transformer architectures with a
specific focus on the non-linearity present in their MLP blocks". How about the non-linear operation inside the attention block, e.g., softmax?

2. In Figure 2, the authors highlight the robustness of the affinity score. Nevertheless, it is still unclear why the robustness of this score is important.

3. It is unclear how accurate the affinity score is to evalute the non-linearity. More details are required towards this.

4. It seems that the non-linearity does not vecessarily correlate with the performance. From this point of view, how do we understand DNNs based on this metric?

**Questions:**

Please refer to the weakness part.

---

> ### Author Response · Authors · 2023-11-16
> **Clearing out misunderstandings**
>
> > 1. The authors mention that they "consider transformer architectures with a specific focus on the non-linearity present in their MLP blocks". How about the non-linear operation inside the attention block, e.g., softmax?
>
> As such, we can consider any transformation and measure its non-linearity. However, we have chosen the activation functions as they are present in all studied models, often repeatedly. We can include pooling layers as well if we were to extend this work. However, we believe that activation functions, often appearing after conv and pool layers are more interesting in this regard as they effectively capture the non-linearity of what precedes them.
>
> Finally, we note that pooling layers cannot be fully linear: it is not clear what an upper bound of the maxpooling operation may look like contrary to activation functions that, in theory, can all attain values between 0 and 1.
>
> > 2. In Figure 2, the authors highlight the robustness of the affinity score. Nevertheless, it is still unclear why the robustness of this score is important.
>
> Neural networks operate on 4th-order tensors. We represent them as 2D matrices and want to know whether such a transformation doesn't lose much information. Figure 2 gives an affirmative answer to this: affinity score is robust to this change of representation.
>
> > 3. It is unclear how accurate the affinity score is to evaluate the non-linearity. More details are required for this.
>
> Our work lays the theoretical foundation in this sense. Note that there are no theoretically sound competitors to our work that can be used as a baseline. Nevertheless, we refer the reviewer to our study of different activation functions (in Figure 1 and more detailed in Appendix B), where we can clearly see that the resulting affinity score is high in regions of the space where the function is linear and low otherwise.
>
> > 4. It seems that the non-linearity does not necessarily correlate with the performance. From this point of view, how do we understand DNNs based on this metric?
>
> The results provided in Figure 9 and Appendix I tell us that for many models this is the case. Can the reviewer be more specific about what gave them this impression?

---

### Official Review · Reviewer_uebp · 2023-10-31

**Soundness:** 2 fair
**Presentation:** 2 fair
**Contribution:** 3 good
**Rating:** 5
**Confidence:** 4

**Summary:**

The paper studies various different NN architectures through a a unique metric of non-linearity. It shows how different networks throughout history have leveraged non-linearity and how they improved. It also shows some theoretical guarantees.

**Strengths:**

The paper is quite interesting! I think it goes through various architectures and shows some compelling results.

**Weaknesses:**

The presentation can be greatly improved in my opinion. The figures are not clear, colormaps are missing, many effects in the plots are not thoroughly explained (e.g. the effects in most figures except the ViTs are not very clear). I think this paper has the potential to be much better if the writing were to be improved, along with the presentation, and more thorough explanations.

**Questions:**

Minor comments:
1. Plots are not clear, they do not have a color bar, and the colors are not clear (e.g. Figure 4, 5, ...)

Questions:
1. Have the authors tried to check how the non-linearity metric corresponds with intermediate layer performance (e.g. the linear eval on the layer)? This could be interesting to check to improve the paper.

---

> ### Author Response · Authors · 2023-11-16
> **Clearing out misunderstandings**
>
> > The presentation can be greatly improved in my opinion. The figures are not clear, colormaps are missing, and many effects in the plots are not thoroughly explained (e.g. the effects in most figures except the ViTs are not very clear). I think this paper has the potential to be much better if the writing were to be improved, along with the presentation, and more thorough explanations.
>
> We would be grateful if the reviewer could provide us with anything specific regarding their last remark. We are wondering which effects in the figures the reviewer is referring to since we analyze each set of figures in the associated text.
>
> Also, please note that this color palette was initially chosen to be friendly for color-blind people. We propose an alternative revised version taking into account your remark.
> When the colormaps are removed, it usually is because they are of no importance: as mentioned in the paper, each color simply is associated with each repeated activation function in the network.
>
> > Have the authors tried to check how the non-linearity metric corresponds with intermediate layer performance (e.g. the linear eval on the layer)? This could be interesting to check to improve the paper.
>
> This is an interesting suggestion, although we do not immediately see what kind of link the reviewer expects to see here. Can they elaborate on their intuition?

---

### Official Review · Reviewer_K2zC · 2023-11-03

**Soundness:** 2 fair
**Presentation:** 3 good
**Contribution:** 3 good
**Rating:** 5
**Confidence:** 2

**Summary:**

This paper proposes a formulation, named 'affinity score,' to measure the non-linearity of deep neural networks. The authors then use the proposed affinity score to evaluate the non-linearity signatures of popular neural networks and illustrate the affinity scores for every layer within these networks. The authors claim that these non-linearity scores will bring insights into the understanding of neural networks.

**Strengths:**

1. The topic of understanding non-linearity in DNNs is intriguing, and the author proposes a novel formulation to evaluate it.
2. The writing style is engaging, and the presentation of the material is well-executed.

**Weaknesses:**

1. In the computation of affinity scores, which are tied to the activations within a neural network, there is an inherent dependence on both the input data and the network's parameters. My understanding is that affinity scores are influenced by the architecture and the parameters (the trained weights) of the network. However, it seems that in the evaluation of their experiments, the authors have placed a predominant focus on the architecture while possibly overlooking the significance of the network parameters. It is important to consider that the parameters, which are shaped by the training process, are crucial for the network's performance and ultimately for the validity of the affinity scores. An in-depth analysis that includes the impact of these parameters could provide a more comprehensive understanding of the network's behavior and the experimental outcomes.
2. The author claims "Despite being almost 20 times smaller than VGG16, the accuracy of Googlenet on Imagenet remains comparable, suggesting that increasing and varying the linearity is a way to have high accuracy with a limited computational complexity compared to predecessors.”
However, this assertion seems to overlook the fact that a significant portion of VGG16's parameters are concentrated in the final fully connected layers, which consist of two 4096-dimensional layers. Empirical evidence suggests that reducing the parameter count of these fully connected layers does not drastically diminish performance. Thus, concluding that 'increasing and varying the linearity is a way to have high accuracy with a limited computational complexity' may not be entirely justified.

**Questions:**

Please refer to the weaknesses.

---

> ### Author Response · Authors · 2023-11-16
> **Clearing out misunderstandings**
>
> > 1. In the computation of affinity scores, which are tied to the activations within a neural network, there is an inherent dependence on both the input data and the network's parameters. My understanding is that affinity scores are influenced by the architecture and the parameters (the trained weights) of the network. However, it seems that in the evaluation of their experiments, the authors have placed a predominant focus on the architecture while possibly overlooking the significance of the network parameters. It is important to consider that the parameters, which are shaped by the training process, are crucial for the network's performance and ultimately for the validity of the affinity scores. An in-depth analysis that includes the impact of these parameters could provide a more comprehensive understanding of the network's behavior and the experimental outcomes.
>
> Thank you for pointing this out so that we can clarify this potential misunderstanding. We actually do not put any particular emphasis on the architecture when defining the non-linearity of the neural network vs. the weights (parameters) of the model. To understand this, one needs to remember that the non-linearity of the activation function reflects what happens in the layers that precede it. As such, ReLU, for instance, is parameterless but the layers preceding it make it more or less linear depending on how they transform the input of the ReLU. These transformations are done by the parameters of the preceeding layers and affinit score reflects this.
>
> Also, please note that the experiments in our paper already cover models trained on several datasets (different sets of parameters, pre-trained or trained from scratch, Appendix F and E) and even different training approaches and learning tasks (see a comparison of self-supervised approaches in Appendix G). We also provide a comparison of models trained with 9 different random seeds (Appendix C). This totals a study over a very large ensemble of possible scenarios.
>
> All in all, we have carried out a comprehensive set of comparisons; it does not seem straightforward what other in-depth analyses would help more. It would be kind of you to detail more in case you had some specific analyses in mind.
>
> > 2. The author claims "Despite being almost 20 times smaller than VGG16, the accuracy of Googlenet on Imagenet remains comparable, suggesting that increasing and varying the linearity is a way to have high accuracy with a limited computational complexity compared to predecessors.” However, this assertion seems to overlook the fact that a significant portion of VGG16's parameters are concentrated in the final fully connected layers, which consist of two 4096-dimensional layers. Empirical evidence suggests that reducing the parameter count of these fully connected layers does not drastically diminish performance. Thus, concluding that 'increasing and varying the linearity is a way to have high accuracy with a limited computational complexity' may not be entirely justified.
>
> We thank the reviewer for pointing this out to us. However, we see a distinct pattern of GoogleNet that we describe as "increasing and varying the linearity" that VGG (with or without high-dimensional classification layers) doesn't have. We will remove the claim about the size though as the reviewer suggests.

---

### Author Response · Authors · 2023-11-16
**Lack of specific criticism**

We'd like to thank the reviewers for reading our proposal. They all seem to find our work *intriguing* and *interesting*, but this appreciation is not reflected in their evaluation.

We regret the absence of comments related to the core of our scientific contribution, which lies in the construction of the first theoretically sound tool for measuring the nonlinearity of a function, or to the extensive experimental study of all the popular deep models proposed over the last decade through the prism of their nonlinearity.

We would like to emphasize that this is the first contribution of its kind, so there is no "benchmark" to compare with or "validation" to perform.

With this in mind, we would greatly appreciate it if the evaluators could specify the major reasons justifying such an assessment, enabling us to engage with them in a substantive scientific discussion.

---

### Meta-Review · Area_Chair_BVFW · 2023-12-06

**Metareview:**

The paper presents a new metric, namely affinity score, to quantify the non-linearity of activation function in DNN. Reviewers appreciate the contributions but raise a few concerns, including the influence of trainable parameters on affinity scores, the limitation of computing scores within the pointwise activation function, and the practical implications of affinity scores in enhancing DNNs. In the rebuttal phase, the authors have made a good effort to clarify the relations between scores and model weights, as well as the potential extension to measure scores for more general activation functions. However, a primary concern persists regarding the practical applications and implications of the affinity score metric. Specifically, it remains unclear whether this metric can be effectively employed to design improved large-scale DNNs or refine activation functions based on the insights derived from this study. This paper would benefit more from a thorough discussion regarding the practical utility of affinity scores in designing large DNNs.

**Justification For Why Not Higher Score:**

The paper did not receive a higher score primarily due to uncertainties about the practical applications and implications of the affinity score metric. There's a lack of clarity on whether this metric can effectively aid in designing improved large-scale DNNs or refining activation functions based on the study's insights.

**Justification For Why Not Lower Score:**

N/A

---

### Decision · Program_Chairs · 2024-01-16

Reject